# Definition of the Binding Architecture to a Target Promoter of HP1043, the Essential Master Regulator of *Helicobacter pylori*

**DOI:** 10.3390/ijms22157848

**Published:** 2021-07-22

**Authors:** Annamaria Zannoni, Simone Pelliciari, Francesco Musiani, Federica Chiappori, Davide Roncarati, Vincenzo Scarlato

**Affiliations:** 1Department of Pharmacy and Biotechnology (FaBiT), University of Bologna, 40126 Bologna, Italy; annamaria.zannoni2@unibo.it (A.Z.); simone.pelliciari@newcastle.ac.uk (S.P.); francesco.musiani@unibo.it (F.M.); 2Istituto di Tecnologie Biomediche-Consiglio Nazionale delle Ricerche (ITB-CNR), 20054 Segrate, Italy; federica.chiappori@itb.cnr.it

**Keywords:** transcriptional regulator, DNA-protein interaction, *Helicobacter pylori*

## Abstract

HP1043 is an essential orphan response regulator of *Helicobacter pylori* orchestrating multiple crucial cellular processes. Classified as a member of the OmpR/PhoB family of two-component systems, HP1043 exhibits a highly degenerate receiver domain and evolved to function independently of phosphorylation. Here, we investigated the HP1043 binding mode to a target sequence in the *hp1227* promoter (*P_hp1227_*). Scanning mutagenesis of HP1043 DNA-binding domain and consensus sequence led to the identification of residues relevant for the interaction of the protein with a target DNA. These determinants were used as restraints to guide a data-driven protein-DNA docking. Results suggested that, differently from most other response regulators of the same family, HP1043 binds in a head-to-head conformation to the *P_hp1227_* target promoter. HP1043 interacts with DNA largely through charged residues and contacts with both major and minor grooves of the DNA are required for a stable binding. Computational alanine scanning on molecular dynamics trajectory was performed to corroborate our findings. Additionally, in vitro transcription assays confirmed that HP1043 positively stimulates the activity of RNA polymerase.

## 1. Introduction

*Helicobacter pylori* is a widespread human pathogen recognized as a class I carcinogen by the World Health Organization. It represents the primary cause of severe gastrointestinal diseases such as peptic ulcer, gastric adenocarcinoma, and MALT lymphoma [1,2]. The ability of this bacterium to colonize the harsh niche of the stomach and to establish a persistent infection depends on the coordinated expression of several genes, including virulence factors. These factors allow the pathogen to adapt to the environmental conditions encountered during infection and to counteract host-defense mechanisms. A peculiarity of the *H. pylori* bacterium is its small-sized genome (1.66 Mb) that codes for a limited number of regulatory genes. Indeed, only 17 transcriptional regulators out of approximately 1500 predicted open reading frames (ORFs) have been identified and partially characterized [3]. Among these, it is worth mentioning only 3σ factors (the housekeeping σ^80^ and the alternative RNA polymerase sigma subunits σ^54^/σ^28^, both involved in the transcription of flagellar genes), 4 repressors of transcription controlling metal homeostasis (Fur and NikR) or stress response (HrcA and HspR), and several two-component systems. These systems are composed of a histidine kinase and a response regulator. Upon perception of an external signal the sensor kinase catalyzes its auto-phosphorylation and then transfers the phosphoryl group to the partner response regulator, a specific DNA binding protein that modulates transcription of target genes.

*H. pylori* adopts two-component systems to coordinate gene expression in important cellular processes such as chemotaxis (CheA/CheY) [4], copper resistance (CrdS/CrdR) [5], flagellar regulation (FlgS/FlgR) [6], and acid acclimation (ArsS/ArsR) [7]. To this class of regulators belong two genes, named *hp1043* and *hp1021*, encoding two response regulators missing their partner sensor kinases. Intriguingly, *hp1043* appears to be essential for cell viability, as it could not be deleted unless a second gene copy was integrated into the *H. pylori* chromosome [8,9], but also the amount of protein could not be modulated by the addition of an extra copy of the wild-type gene under the control of an inducible or strong constitutive promoter [10,11], nor by interference with an antisense RNA [12], suggesting a tight regulation of its expression. The impossibility to modulate HP1043 protein levels severely hampered the possibility to study the mechanisms of its regulation. However, by using in vivo chromatin immunoprecipitation-sequencing (ChIP-seq), a direct regulon was determined [13]. This analysis, likely not exhaustive, because of a stringent peak-calling analysis applied to identify high confidence candidates, identified 37 highly reproducible binding sites accounting for a total of 97 genes involved in several fundamental processes in the cell, including translation, transcription, replication, energy metabolism, and virulence [13]. In the same study, alignment of experimentally validated target sequences was used to define the consensus binding sequence of HP1043, which consists of a repetition of two TTTAAG hexamers spaced by 5 bp and overlapping promoter sequences [13]. This positioning is typical of transcriptional activators [14].

The structure of full-length HP1043 solved by NMR [15,16] showed that its molecular topology resembles that of the OmpR/PhoB family [16]. HP1043 exists as a 2-fold symmetric dimer in solution in the absence of phosphorylation and has two functional domains: an N-terminal dimerization domain (HP1043N, residues 1–113) and a C-terminal DNA binding domain (HP1043_DBD, residues 121–223), connected by a flexible linker (residues 114–120). Different biochemical and structural studies suggested that HP1043 could exert its function in a phosphorylation-independent manner [9,11].

In the present study, we set up biochemical and computational approaches to define the binding architecture of the HP1043 regulator to a selected target promoter, *P_hp1227_*. We applied nucleotide scanning mutagenesis to the consensus binding sequence of HP1043 to identify relevant nucleotides for efficient binding, highlighting an asymmetry in recognition of the two hemi-sites by HP1043. Furthermore, alanine scanning mutagenesis of the HP1043_DBD led to the identification of amino acids fundamental for DNA binding. These experimental data, in combination with the available HP1043 NMR structure, were used as restraints to guide an in-silico protein-DNA docking. The generated model shows an HP1043 dimer interacting in a head-to-head conformation with both the major and minor groove of a target DNA sequence. Although this configuration differs from the canonical head-to-tail conformation of other regulators of the OmpR/PhoB family, site-directed mutagenesis and in vitro binding assays validated the model. Additionally, in vitro transcription assays performed to evaluate the effect of HP1043 on the activity of the RNA polymerase confirmed that HP1043 acts as a transcriptional activator, at least on the *P_hp1227_* promoter.

## 2. Results

### 2.1. Molecular Characterization of HP1043 Consensus Binding Sequence

The alignment of experimentally validated HP1043 binding sites led to the definition of a consensus sequence composed by the repetition of two TTTAAG hexamers separated by a 5 bp spacer, with a lower degree of sequence conservation for the second hemi-site [13]. To investigate the relevance of single nucleotides of the consensus for recognition of a target sequence by HP1043, we adopted a scanning mutagenesis approach. A DNA probe encompassing the HP1043 binding site on the *hp1227* promoter (*P_hp1227_*) was selected as a model for mutagenesis, due to the high binding affinity of HP1043 for this promoter in in-vitro experiments and the elevated degree of conservation to the HP1043 consensus sequence (TTTAAG-5bp-CTTAAA) [13]. Each nucleotide of the consensus was substituted with the complementary base to avoid altering the overall GC content of the probes, and HP1043 binding to mutated targets was tested in high-resolution DNase I footprinting experiments (Figure 1).

As previously reported, binding of HP1043 to the wild-type *P_hp1227_* generates a clear region of protection from DNase I digestion (footprint) that spans nucleotides from −23 to −42 relative to the *hp1227* transcriptional start site (TSS), already at the lower protein concentration tested (1.7 µM), with the consensus sequence spanning from position −26 to position −42 (Figure 1A,B, WT). Two sites of enhanced sensitivity to DNase I digestion were also identified just outside the extremities of the binding site, one mapping to position −48, and one mapping to position −15, indicating a distortion of the DNA double helix upon HP1043 binding.

In regard to the first hemi-site of the consensus (Figure 1A), mutation of the first nucleotide, named T1F, did not alter HP1043 binding affinity for the target DNA, as protection of the probe appeared at the lower amount of HP1043 protein tested as in the wild-type (Figure 1B). T2F and T3F mutants exhibited slightly altered binding patterns, and protection by HP1043 appeared at a higher protein concentration (6.6 µM). Incubation of the protein with A1F, A2F, and GF mutant probes showed a diminished area of protection, centered on the second hemi-site, indicating a weaker binding by HP1043. In the case of mutants A1F and A2F, complete protection of the bound region was not yielded even at the highest protein concentration (13.3 µM), highlighting the relevance of these nucleotides for efficient binding by HP1043. Notably, the DNase I hypersensitive site located upstream the first hemi-site of the consensus weakened in A1F, A2F, and GF mutants, in agreement with the apparent reduced affinity of HP1043 for these mutant probes. Considering the second hemi-site, mutants T1S and GS (Figure 1C,D) showed a digestion pattern similar to the wild-type probe, whereas mutation of the four central nucleotides impaired HP1043 binding to the target DNA. In particular, T3S and A2S showed a footprint only at the highest protein concentration, while T2S and A1S mutants almost completely abolished protection from DNase I digestion. Taken together, these results evidence an asymmetry in recognition of the two hemi-sites of the consensus sequence by HP1043, where nucleotides A1, A2, and G of the first hemi-site and nucleotides T2, T3, A1, and A2 of the second hemi-site play a pivotal role for recognition of a target DNA by HP1043.

Moreover, as HP1043 forms dimers in solution, we tested whether HP1043 could interact with a DNA binding site lacking one of the TTTAAG hexamers. Incubation of the protein with mutant probes in which the entire first or second hemi-site was substituted with a non-related nucleotide sequence (mutants ΔF and ΔS, Figure 1) determined a complete loss of protection in vitro, demonstrating that a single hemi-site of the consensus is not sufficient for stable binding of the protein to a target DNA. Additional DNase I hypersensitive sites detected at high HP1043 protein concentrations in ΔF and ΔS footprinting experiments (marked with arrows in Figure 1B,D) probably represent non-specific contacts of HP1043 to these mutant probes.

To further validate these results and evaluate the extent to which different mutations of the consensus might affect the HP1043 role in the process of transcription, we performed in vitro transcription assays using the *E. coli* RNA polymerase holoenzyme (RNAP) and a plasmid containing the *lux* operon under the control of *P_hp1227_* as a template in the presence/absence of HP1043. The amount of synthesized transcripts was quantified by qRT-PCR. In the absence of HP1043, transcription from mutant promoters did not show any significant variation from the wild type, indicating that promoter recognition and transcription by the RNAP were not affected by the introduced mutations in the HP1043 consensus binding sequence (Figure 1E). In the presence of a 3:1 molar ratio of HP1043 to RNA polymerase, a 5-fold increase of transcripts was obtained for the wild-type *P_hp1227_* compared to basal levels, supporting a role of HP1043 as a transcriptional activator (Figure 1F, first column marked as WT). This result is coherent with a previous observation that genes bound in vivo by HP1043 were upregulated in response to translational arrest [13]. In the same experimental conditions, all the tested mutant promoters exhibited a minor increase in the amounts of transcripts compared to the wild type, demonstrating that deviation from the consensus affects the efficiency of transcriptional stimulation by HP1043 (Figure 1F). As expected, mutations of the probe that we found to hamper significantly HP1043 binding in vitro (A1F, A2F, GF, T2S, T3S, A1S, A2S, ΔF, and ΔS) also showed a level of transcription similar to the control experiments, indicating that the addition of the regulator had no effects on transcription. Interestingly, also the T3F mutation prevented transcriptional activation by HP1043, while the addition of HP1043 to the GF mutant reaction caused a decrease in transcript levels. This latter observation, although not further investigated, could be explained as an altered HP1043 binding to the GF mutant promoter which interferes with the activity of the RNA polymerase. In support of this hypothesis is the appearance of a novel hypertensive site to DNase I digestion in position −40 of the GF mutant probe, close to the −35 promoter element (Figure 1B, mutant GF). Overall, these findings corroborate our previous in vitro binding assays assessing HP1043 interaction with WT and mutant probes by DNase I footprinting and suggest that a proper binding of HP1043 to its target is required to stimulate transcription by RNAP. In light of the results described above, a consensus based on the relevance of nucleotides in the HP1043 binding sequence might therefore be written as ttTAAG-5bp-tTTAAg, where capital letters represent the most important bases. Notably, it is worth highlighting that the consensus sequence for HP1043 binding analyzed in the *P_hp1227_* promoter can be interpreted as a direct as well as an inverted repeat.

### 2.2. Identification of Amino Acid Residues Fundamental for Target DNA Recognition by HP1043

Structural data of the HP1043 protein revealed that its DNA binding domain (HP1043_DBD) is characterized by a winged-helix-turn-helix (wHTH) motif, where the helices α7 and α8 form the HTH motif and the strands β11 and β12 flank the winged loop [16,17] (Figure 2A). Helix α8 likely represents the recognition helix of the motif, interacting with the major groove of target DNA, while residues included in the β11–β12 winged loop have been proposed to contribute to DNA binding [16].

To assess which residues of the HP1043 wHTH motif are fundamental for the recognition of target promoters, we exploited the alanine scanning mutagenesis approach. One of the criteria that guided our selection was the evolutionary conservation rate of the residues to estimate their biological importance, as amino acids that are critical for maintaining the structure of a protein and/or its function often undergo evolutionary constraints and are, therefore, invariant within the protein family. To help determine the relevance of single HP1043 residues for DNA recognition we exploited ConSurf, a web server that computes evolutionary conservation scores for each amino acid position and maps them onto the 3D structure of the protein [18]. As expected, the majority of the residues belonging to the wHTH motif are highly conserved, exception made for K197, supporting their relevance for the functionality of HP1043 (Figure 2B,D).

As a second criterium, we evaluated the spatial orientation of the residues belonging to helix α8 and the β11–β12 winged loop in the HP1043 NMR structure that appeared favorable for interaction with DNA. Following these criteria, we selected residues I188, N189, R192, Q193, D196, and K197 of helix α8 and amino acids T206 and R208 of β11–β12 winged loop (Figure 2C). Notably, a few of these residues (N189, Q193, T206, and R208) were already reported to affect HP1043 binding in vitro and amino acids N189 and T206 were found among those showing the highest chemical shift perturbation in an NMR titration experiment with the consensus DNA [16]. Recombinant wild-type and mutant HP1043 proteins were then assessed in DNase I footprinting assays for their ability to bind in vitro to the *P_hp1227_* wild-type probe (Figure 2E). In our experimental settings, mutation of residues I188, R192, D196, and K197 in the recognition helix and of residues T206 and R208 in the β-hairpin wing completely abolished HP1043 binding, since effective DNA protection from DNase I digestion was not achieved even at a protein concentration 27-times higher than that required for partial protection of the probe by the wild-type protein. The result suggested that these residues are fundamental for the recognition and stable binding of target DNA by HP1043. However, I188A and D196A mutations reduced the solubility of the protein preparations and hampered the stability in thermal shift assays (Figure 3D), suggesting that these residues might be critical for the correct folding of the HP1043_DBD. For this reason, residues I188 and D196 were excluded from further analysis. Considering the other selected residues, T206D and R208A mutants were previously reported to abolish HP1043 binding to DNA in vitro, confirming our result [16], while the relevance of R192 and K197 was first estimated in this study. Interestingly, K197 is the only residue in the recognition helix with a low evolutionary conservation score (Figure 2B,D), suggesting that this HP1043 residue could be an element of discrimination from orthologues for the recognition of their targets. By contrast, in previous research, mutations N189A and Q193G in the HP1043_DBD resulted in decreased DNA binding activity in vitro [16]. However, in our experimental setup, N189A and Q193A mutants showed a DNA binding affinity comparable to the wild type and therefore were not further investigated (Figure 2E).

In addition, we investigated whether dimerization of HP1043 is required for interaction with DNA. To this aim, we tested an HP1043 mutant encompassing its DBD in in-vitro binding experiments (Figure 2E, CTD mutant). This mutant was not able to protect the target promoter in DNase I footprinting experiments, likely indicating that dimerization is a prerequisite for HP1043 stable binding to DNA.

### 2.3. Structural Modeling of the HP1043_DBD-DNA Interaction

In the absence of protein-DNA co-crystals, we took advantage of the experimental information gained in this study for the identification of the most relevant residues in both the *P_hp1227_* target promoter and the HP1043_DBD to guide a two-stage data-driven docking with the program HADDOCK 2.2 [19,20] and produce a computational model of the HP1043-DNA interaction. In particular, nucleotides tttttTAAGcaaaacTTAAacttgta (in uppercase) and protein residues R192, K197, T206, and R208 were used as initial molecular restraints to guide the interaction. The selected nucleotides and amino-acid residues are the ones that proved to be fundamental for DNA binding in vitro. Due to the flexibility of the HP1043 interdomain linker, the computational analysis was performed on a library of 20 HP1043_DBDs derived from the available NMR structure of the full-length protein (PDB: 2HQR), by imposing interaction of one HP1043_DBD with the selected nucleotides of the first hemi-site of the consensus sequence, and of another HP1043_DBD with the residues selected from the second hemi-site. Results of the first docking round were manually screened, to select clusters in which the relative orientation of the HP1043_DBDs was compatible with the distances allowed by the linker between the DBD and the dimeric N-terminal domain. Interestingly, the only cluster that fitted this requirement showed a 180° rotation of the DBDs compared to the conformation of the dimeric HP1043 in solution, and this configuration appeared to be stabilized by reciprocal interaction between residues E143, V144, K145 at the interface of the two DBDs (Figure 3A,B). This cluster was selected to perform a second docking run (including the same restraints of the previous run plus those linking E143 and K145 from each HP1043_DBD monomer with their counterpart on the other monomer) and obtain the final model (Figure 3A). In our model, the HP1043_DBDs interact with DNA both at the major and the minor groove through helices α8 and wings β11–β12, respectively, although the constraints used in the docking algorithm were not explicitly set to reward interactions with the minor groove. Moreover, the interaction with the major groove of helices α8 with the two hexamers of the consensus binding sequence determines a narrowing of three adjacent minor grooves. This is in agreement with previous hydroxyl-radical footprinting experiments where three areas of protection were found on target probes, reflecting limited access to radical ions cleavages upon HP1043 binding [13]. In the model, the two HP1043_DBDs are bound on the same side of the DNA molecule, in congruence with the 1-helix turn that distances the center of the two hexamers of the consensus binding sequence.

The modeled HP1043_DBDs are oriented in a head-to-head conformation which would explain the minor conservation of the second hemi-site of the HP1043 consensus sequence and the asymmetry in recognition of the two hemi-sites by HP1043 in our in vitro experiments. This modeled configuration of HP1043 differs from that of other dimeric regulators of the OmpR/PhoB family such as *E. coli* PhoB, and *Klebsiella pneumoniae* RstA and PmrA for which the sole DBDs or the full-length protein were co-crystalized with DNA in a head-to-tail conformation, coherently with the recognition of a direct repeat [21,22,23]. Nonetheless, OmpR can recognize different DNA sequences and bind in either a head-to-tail or head-to-head orientation [24,25,26]. Such diversities in the way in which regulators of the same protein subfamily recognize target tandem DNA sequences were proposed to depend on the length and flexibility of the interdomain linker [27]. However relatively short (residues 114–120), the presence of a Glycine residue (G117) at the center of the flexible linker could allow otherwise forbidden rotation angles.

### 2.4. Validation of the HP1043-P_hp1227_ Docking Model

To start the experimental validation of our computational model, we selected, for each of the functional elements of the wHTH motif a couple of amino acid residues, one of which is expected from the inspection of the model to interact with the DNA and another amino acid which appeared not to be involved in the interaction to be tested by alanine scanning mutagenesis. In detail, residues K194 and Q190 for the α8 helix, and residues R210 and R209 in the β11–β12 winged loop were selected, respectively (Figure 3A). In agreement with the model, mutation of residues K194 and R210 partially (for K194) or completely (for R210) impaired protection of the probe by HP1043 in in vitro binding assays, while the proteins harboring mutations Q190 and R209 retained the ability to bind DNA (Figure 3C). Furthermore, to validate the putative EVK interaction at the interface of the DBDs, multiple mutants were produced (Figure 3B). The HP1043 mutant in which the EVK triplet was converted to AAA was not able to protect the target DNA in footprinting experiments. However, this mutation altered the thermal stability of the protein and was therefore excluded from further analyses (Figure 3D). In contrast, the EAA and AAE mutants lost their binding property, retaining a thermal unfolding profile similar to that of the wild type, although the overall fluorescence signal was lower (Figure 3D). Since the EVK triplet is positioned in the loop connecting β8 and α6 which is exposed to the solvent in the HP1043 configuration resolved by NMR, the results of in vitro binding assays could support an involvement of the triplet in the stabilization of an HP1043 conformation that is functional for interaction with the DNA. However, additional experiments are required to rule out the involvement of these residues in different intra- or inter-domain interactions that might be relevant for the HP1043 function. Additionally, DNA binding interference experiments were implemented to estimate the relative relevance of HP1043 interactions with minor and major grooves of the DNA (Figure 3E–G). In particular, once the HP1043 concentration required to shift the target P*_hp1227_* probe was determined (~8 μM, Figure 3E), in vitro binding reactions were performed in the presence of increasing concentrations of the double-stranded DNA (dsDNA) groove binders Netropsin and Methyl green. Specifically, Netropsin is an oligopeptide that binds non-covalently to the minor groove of dsDNA preferably in AT-rich regions, while Methyl green is a positively charged dye that forms electrostatic interactions with negatively charged phosphate radicals in the major groove of dsDNA. In our experimental settings, Netropsin interfered with HP1043 binding at nanomolar concentrations, and completely abolished protein-dependent specific shift of *P_hp1227_* at a concentration of 1 μM, which is almost one order of magnitude less than the HP1043 concentration, demonstrating the high relevance of HP1043 contacts at the DNA minor groove (Figure 3F). Oppositely, Methyl green did not interfere with HP1043 binding even at a concentration 60-times higher than that of the protein (Appendix A). Since HP1043 was proven to interact extensively with the major groove via the α8 recognition helix of the HTH motif, the result could be explained as a stronger affinity of the protein for the DNA compared to Methyl green. To prove this hypothesis, we exploited the fact that hydrogen bonding on purine and pyrimidine rings in inosine-cytosine I-C pairs resembles A-T pairs in the minor groove and G-C couples in the major groove. Therefore, we tested the ability of HP1043 to bind an *hp1227* promoter probe harboring I-C box substitutions in A-T couples that previously proved to be most important for HP1043 recognition of its consensus binding sequence (Figure 3G). HP1043 failed to bind and shift the I-C box substituted target promoter even at a protein concentration of 24 μM (Figure 3G), therefore we can conclude that in the previous experiment HP1043 out-competed Methyl green for binding to the DNA and that both a major and minor groove readout are required for recognition and binding of target sequences by the regulator.

### 2.5. Conformational and Interaction Analysis of HP1043-DNA Full Model

To reconstitute the full HP1043-DNA model (Figure 4), the HP1043N domain and the flexible linker were added to each HP1043_DBD monomer bound to DNA by adopting a two-step strategy. In the first step, the HP1043N domain in the dimeric form was docked onto the HP1043_DBD/*P_hp1227_* complex obtained previously (see Section 2.4). Considering that the flexible linker is only six residues long, we explored a model able to bring the C-terminals of each HP1043N domain in the vicinity of the N-terminals of the HP1043_DBD domains already bound to the DNA. In agreement with this hypothesis, the two fragments (i.e., the HP1043N domain dimer and the HP1043_DBD/*P_hp1227_* complex) were placed in a guess orientation at a distance of 10 Å and submitted to the software RosettaDock [28,29] to execute the docking and to refine their relative orientations. The resulting HP1043N/HP1043_DBD/*P_hp1227_* complex was then finalized in the second step of the reconstruction procedure and by using the software MODELLER [30]. The results of the docking and modeling procedure are reported in Figure 4. In the modeled HP1043/*P_hp1227_* complex, the HP1043N dimer is rotated by ca. 30° with respect to the HP1043_DBD/*P_hp1227_* major axis (corresponding approximately with the major axis of the DNA double helix). Each HP1043N monomer interacts with the corresponding HP1043_DBD domain through the N-terminal and the initial part of strand β1, strand β2, and the loop connecting helix α3 and strand β3. On the HP1043_DBD side, the interaction is due to strands β6 and β9 and the loop connecting the strands β7 and β8.

To estimate the conformational stability of the HP1043-DNA model and the maintenance of HP1043 interaction with the target DNA, we performed 100 ns of molecular dynamics simulation either on the protein or on the protein-DNA complex (see Section 4.1, Section 4.2, Section 4.3, Section 4.4, Section 4.5, Section 4.6, Section 4.7, Section 4.8, Section 4.9 and Section 4.10 for details). We used Normal mode analysis to evaluate the motion direction of the DBDs, which display an increasing reciprocal distance during the simulation. Figure 5 shows that the movement of the DNA-bound and unbound conformations is similar. Comparable results were obtained with cluster analysis of the conformational sampling, where 4 clusters for the DNA-bound and 3 for the unbound HP1043 were obtained (Appendix A). HP1043 displays major mobility of the DBD of chain A (residues 113–223) compared to chain B in both clustering conformations. Moreover, to evaluate the protein-DNA complex conformational stability and whether the two chains display a different binding strength for the DNA, the distance between DBD residues and DNA was measured along the simulations. In particular, the distances between residues I188, N189, R192, Q193, D196, K197, T206, R208 and the DNA center of mass, as well as the distance between the DBD domain of each chain and the DNA centers of mass, were measured (Appendix A). Distances of both domains appear stable along the trajectories, although residues belonging to chain A display a higher distance from DNA. This is probably due to the higher mobility of this domain, or the different positioning of the two chains on the DNA molecule. Specifically, chain A interacts with the first hexamer of the HP1043 consensus, which is very lateral on the modeled DNA molecule and therefore more mobile, in contrast to the second hexamer recognized by chain B, located closer to the center of the DNA molecule. Also, the shorter distance of chain B residues allowed us to hypothesize a greater binding strength. Despite the observation of the DBDs separation, residues at the protein-DNA interface were maintained compared to the docking model. In particular, considering the mean conformation obtained from the most populated cluster, residues N189, R192, Q193, D196, T206, and T208 compose the protein interaction surface for DNA in both chains. By contrast, residues I188, Q190, K194, and K197 are included in chain B only.

To better characterize the HP1043 residues fundamental for DNA binding, we performed an alanine scanning analysis on the HP1043-DNA equilibrated trajectory employing the MMGBSA method, which was previously evidenced to be a reliable method for the study of nucleic acids [31]. This analysis allows us to estimate the change in terms of binding energy for the DNA molecule upon substitution of each considered residue (Figure 6) to alanine. For residues I188, N189, Q190, D196, K197, and V207A, alanine scanning analysis returned similar results for the two chains, while for residues R192, Q103, K194, T206, R208, and R209 alanine substitution appears to have a greater impact on the DNA-binding energy calculated for chain B (Appendix A). This correlates with the shorter distance of chain B DBD residues from DNA and the consequent greater binding strength of this domain that we postulated. Residues I188 and D196 are neutral to alanine substitution, while residues R192, Q193, K194, and R208 display relevant differences in binding energy. Alanine substitution of residues N189, Q190, K197, T206, and R209 show a minor effect on the binding capacity to the DNA, that for residues T206 and R209 is greater in chain B. Considering the mean conformation obtained from the most populated cluster, both neutral residues localize far from the DNA. Instead, residues R192, Q193, K194, and R208 display side chains towards the DNA, mainly to the backbone, and among these, R208 localizes in the minor groove. The remaining residues are in close contact with residues directly involved in DNA interaction. Overall, residues R192, K194, T206, R208, and R209 that were experimentally proven in this study to hamper or abolish DNA binding were also evidenced as relevant for DNA interaction by computational alanine scanning. Notably, the analysis also highlighted a possible involvement of residue Q193 in DNA binding, as previously reported [16].

## 3. Discussion

In this study, we aimed at characterizing the molecular mechanism of interaction between HP1043 and its target DNA, and its role as a transcriptional regulator. In vitro transcription assays were performed to evaluate the effect of HP1043 on the activity of RNA polymerase (RNAP). The addition of HP1043 to the reaction determined a 5-fold increment in mRNA levels, supporting its role as a transcriptional activator. To correlate the binding of HP1043 to the increment in mRNA level generated by RNAP, the same in vitro transcription experiment was performed with *P_hp1227_* probes harboring mutations in the HP1043 consensus binding sequence. *P_hp1227_* mutants that were proven to affect HP1043 binding in vitro failed to increase transcription rates. Thus, our results prove that, at least on the *P_hp1227_* promoter, HP1043 binding is responsible for the stimulation of RNAP and transcription increase.

In the absence of co-crystals, we exploited in vitro binding, transcription assays, and scanning mutagenesis approaches to identify the most relevant residues in both the DNA and the HP1043_DBD for efficient binding of the regulator to a target promoter (*P_hp1227_*). The different degrees of nucleotide conservation between the two hexamers of the HP1043 consensus binding sequence suggested that the first repeated sequence might play a primary role in HP1043 binding [13]. However, in our experimental conditions, deletion of either of the two hemi-sites of the consensus abolished HP1043 binding to the target DNA in vitro, demonstrating that both hemi-sites are required for stable binding of the protein to a target DNA. Experiments of scanning mutagenesis and in vitro transcription assays showed that sequence conservation is not the only parameter to consider for defining the relevance of each nucleotide in a target DNA for recognition by HP1043. In particular, in the case of the first hemi-site, mutation of nucleotides T3F, A1F, A2F and GF mostly compromised binding of HP1043 to DNA and transcriptional activation from a target promoter in vitro. In the second hemi-site, although less conserved, mutation of 4 out of 6 nucleotides (T2S, T3S, A1S, A2S) hampered recognition by HP1043 (Figure 1). A consensus based on the relevance of nucleotides in the HP1043 binding sequence might therefore be written as ttTAAG-5bp-tTTAAg, where capital letters represent the most important bases. These results also evidenced an asymmetry in the mechanism of recognition of the two hemi-sites by HP1043, compatible with a different orientation of the two HP1043-DBDs on the DNA. On the other hand, mutational analysis of the DNA binding domain led to the identification of a few charged residues (R192, K197, T206, and R208) that are fundamental for the specific interaction with DNA. Experimentally identified residues in DNA and HP1043_DBD most important for interaction, in combination with the available HP1043 NMR structure, were used as restraints to guide an in-silico protein-DNA docking. The generated model showed an HP1043 dimer interacting in a head-to-head conformation with both the major and minor groove of a target DNA sequence. Site-directed mutagenesis and in vitro binding assays validated the model. Computational alanine scanning was performed to identify the energetically important residues at the protein-DNA interface and further supported experimental results.

*H. pylori* is one of the most widespread and successful human pathogens. Although only approximately 1% of *H. pylori* carriers is expected to develop associated malignancies, the worldwide dissemination, high prevalence, and life-long persistence of the infection, together with the increasing emergence of antibiotic-resistant strains pinpointed the necessity to develop novel antimicrobial strategies to eradicate the bacterium [32]. A trending strategy in the field is represented by the targeting of essential bacterial transcriptional regulators (BTRs) [33], allowing to obtain both a specific and multitargeting effect, as BTRs are usually soluble cytoplasmic proteins that lack a human counterpart, and often a single BTR regulates multiple genes. In the case of *H. pylori*, the transcriptional regulator HP1043 represents an ideal target, being a master regulator of all the fundamental processes of the cell and therefore essential for the viability of the bacterium [13], and recent research has been produced on this topic [34,35]. The OmpR/PhoB family is the widest family of bacterial response regulators proteins, therefore the identification of ligands that interact selectively with a specific regulator is mandatory to avoid off-targeting effects and alterations to the microbiota. Since the configuration of the HP1043_DBDs determined through our virtual docking significantly differs from the canonical head-to-tail conformation of other regulators of the OmpR/PhoB family, our model could be used to conduct a virtual screening of potential inhibitors highly specific for HP1043 and therefore *H. pylori*, avoiding potential off-targets. Moreover, the residues identified as fundamental for HP1043 interaction with DNA (R192, K194, K197, T206, R208, and R209) could serve as hot spots to select molecules most promising for inhibiting HP1043 binding to DNA and block its function.

## 4. Materials and Methods

### 4.1. Bacterial Strains and Culture Conditions

*E. coli* DH5α strain [36] was grown in Luria–Bertani (LB) agar or in LB broth (Sigma-Aldrich, St. Louis, MO, USA). When required, media were supplemented with ampicillin (100 μg/mL).

### 4.2. DNA Manipulations

DNA manipulations were performed as described previously [37]. All restriction and modification enzymes were used according to the manufacturers’ instructions (New England Biolabs, Ipswich, MA, USA). Plasmid DNA preparations were carried out with the NucleoBond Xtra Mini/Midi plasmid purification kits (Macherey-Nagel GmbH & Co, Düren, Germany). DNA fragments for cloning purposes were extracted and purified from agarose gel using the NucleoSpin Gel and PCR Clean-up Kit (Macherey-Nagel GmbH & Co., Düren, Germany). PCRs were carried out in a PTC-100 (MJ Research, St. Bruno, QC Canada) or in an AimpliAmp (Applied Biosystems, Waltham, MA, USA) thermal cycler using PCRBIO Classic Taq or HiFi polymerase (PCR Biosystems, London, UK).

### 4.3. Overexpression and Purification of Recombinant His6-HP1043

Recombinant N-terminal His-tagged HP1043 wild-type and mutant proteins were overexpressed in *E. coli* DH5α cells transformed with plasmid pTrc::1043 and its derivatives. For DNase I footprinting assays, HP1043 was affinity-purified as previously described [13] and dialyzed against two changes of 1× 1043 Footprinting Buffer (1× 1043 FPB: 10 mM Tris-HCl pH 7.5; 50 mM NaCl; 10 mM MgCl_2_; 1 mM DTT; 0.01% Igepal CA-630; 10% glycerol). For EMSA assays, HP1043 was purified as in [34]) and dialyzed against the store buffer (50 mM Tris-HCl pH 8, 300 mM NaCl, 10% glycerol). Protein purity was assessed by SDS-PAGE and the concentration of protein preparations was estimated by Bradford colorimetric assay (BioRad, Hercules, CA, USA). HP1043 mutants were obtained by all-around PCR performed on plasmid pTrc::1043 with divergent primers listed in Appendix A.

### 4.4. DNase I Footprinting

Plasmid pGEMt-P1227 WT harboring the HP1043 binding site on the *P_hp1227_* promoter was linearized by enzymatic restriction and 5′-end-labeled by T4 polynucleotide kinase in the presence of [γ32P]-ATP. DNA probes were obtained by second digestion with the appropriate enzyme and gel purified. Approximately 15 fmol of the labeled probe was used per footprinting reaction. DNase I footprinting experiments were performed as previously described [13]. Samples were fractionated on a 6% polyacrylamide—8 M urea sequencing gel in TBE buffer and autoradiographed. A modified G+A sequencing ladder protocol [38] was employed to map the binding sites.

### 4.5. Generation of Probes for DNase I Footprinting and Templates for In Vitro Transcription Experiments

Plasmids containing the DNA probes used in DNase I footprinting experiments were obtained as follows. About 250 pmol oligonucleotides (Appendix A) were phosphorylated with T4 polynucleotide kinase in the presence of 1 mM ATP for 1 hour at 37 °C. Complementary oligonucleotides were denatured for 5 min at 100 °C in 100 mM NaCl and allowed to reanneal at room temperature. Each annealed reaction was cloned into the pBlueScript KSII (+) plasmid (Agilent, Santa Clara, CA, USA) digested with EcoRV and dephosphorylated with Calf Intestinal Phosphatase, generating plasmids PBSK-p1227 or in the pGEMt-Easy plasmid (Agilent, Santa Clara, CA, USA), generating plasmid pGEMt-P1227. Plasmids harboring the luciferase operon under the control of WT and mutant *P_hp1227_* that were used as templates for in vitro transcription reaction were generated by amplifying each promoter from the corresponding PBSK-p1227 plasmid with oligonucleotides p1227EcoRV_F/p1227EcoRV_R, restriction of amplified PCR products with EcoRV, and cloning in pLux plasmid [39] previously digested with the same enzyme and dephosphorylated with Calf Intestinal Phosphatase.

### 4.6. In Vitro Transcription and cDNA Synthesis

For in vitro transcription assays, 6 nM of each pLux plasmid was pre-incubated with 0.2 U of *E. coli* RNA polymerase (New England Biolabs, Ipswich, MA, USA) for 5 min at 37 °C, in a final volume of 16 μL RNA Polymerase Reaction Buffer (New England Biolabs, Ipswich, MA, USA). Increasing concentrations of recombinant purified HP1043 diluted in 1× 1043 FPB were added to the mixture and incubated for 10 min at 37 °C. In vitro transcription was started by the addition of 2 μL of 5 mM NTPs and performed for 10 min at 37 °C. The reactions were stopped with 20 μL of 1M NaCl and subjected to phenol-chloroform extraction and ethanol precipitation. Nucleic acids were resuspended in 12 μL of mqH_2_O and 6 μL of them were treated with 10 U of DNase I in 1× DNase Buffer (Merck Millipore, Darmstadt, Germany) for 1 h at 37 °C to remove plasmid templates. RNA was reverse transcribed to cDNA as previously described [13]. In each cDNA synthesis reaction, 1 ng of *cncr1* RNA produced by in vitro transcription with the MEGAscript T7 Transcription kit (Thermo Fisher Scientific, Waltham, MA, USA) following manufacturer’s instructions was added as an internal control.

### 4.7. qRT-PCR Analysis

For qRT-PCR analyses, 2 μL of the diluted (1:10) cDNA samples were mixed with 5 μL of 2× Power Up SYBR Green Master Mix (ThermoFisher Scientific, Waltham, MA, USA) and 400 nM oligonucleotides (LuxRT_F/Lu × RT_R, Appendix A) in a final volume of 10 μL. Data were analyzed using the ∆∆Ct method. The levels of cDNAs of interest were normalized against the measured level of the cDNA of the *cncr1* gene.

### 4.8. Protein-DNA Docking and HP1043 Structure Reconstruction

The HP1043_DBD-DNA interaction model was obtained using the data-driven docking program HADDOCK 2.2 [19,40] and a previously described protocol [41,42] that involves a two-stage protein-DNA docking approach [20]. The first docking round was performed between a library of HP1043 DNA binding domains derived from the available NMR structure of the whole protein (PDB: 2HQR, residues 120–223) and a linear B-form DNA model of the HP1043 binding site on *P_hp1227_* plus 3 nt on each side (TTTTTTAAGCAAAACTTAAACTTGTA, hexamers of consensus are underlined) generated through the 3DNA software implemented in the 3D-DART server [43]. Amino acid residues of HP1043_DBD (R192, K197, Y206, K208) and nucleotides in *P_hp1227_* (tttttTAAGcaaaacTTAAacttgta, in uppercase) experimentally validated to be most relevant for HP1043-DNA interaction were used as restraints to guide the modeling procedure and are indicated as “active” residues. The docking algorithm rewards the complexes that have “active” protein residues or DNA nucleotides at the interaction interface [19,40]. The second set of “passive” protein residues (K165, I188, N189, Q193, K194, D196, P198, V204, E205, V207, R209, Y212, R213), defined as the residues found on the protein surface and in contact with the active residues, as well as “passive” DNA nucleotides (the same defined as “active”) were included in the calculation. The experimental information is thus translated in the docking process to ambiguous interaction restraints (AIRs) that are used to drive the docking process. An AIR is defined as an ambiguous intermolecular distance with a maximum value of 3 Å between any atom of an active residue of the biomolecule A (HP1043_DBD in the present case) and any atom of both active and passive residues of the biomolecule B (the DNA in the present case) [19,40]. Additional restraints were introduced for the DNA fragment to maintain base planarity and Watson-Crick bonds [20].

Results of the first docking round were manually screened and a second docking round was run with the same setup of the first round plus additional restraints apt to maintain in close contact the two HP1043_DBD monomers in the region of E143 and K145. The 200 models obtained from the second docking run were clustered using a cut-off of 7.5 Å based on the pairwise backbone root mean square deviation matrix. Subsequently, the DNA conformation in the resulting docked structures was analyzed using the program 3D-DART [43] to determine trends in DNA bending and twisting. This information was used to generate an ensemble of custom DNA models representing the accessible conformations, using a local version of the program 3D-DART (https://github.com/haddocking/3D-DART, Version 1.2, February 2018). A subsequent HADDOCK docking round was then carried out following the same approach as described for the second round, but this time including the ensemble of DNA models generated above. In this round, the conformational freedom of the DNA molecule was restricted at the semi-flexible refinement stage to prevent helical deformation. The structure with the best HADDOCK score belonging to the most populated cluster after a new clustering procedure was selected as the best model.

To rebuild the whole HP1043 dimer, the dimeric N-terminal domain of HP1043 (PDB: 2PLN) was docked on the HP1043_DBD/*P_hp1227_* complex using the RosettaDock software [28,29] and adopting a local docking protocol involving the generation of 500 docking poses. The interacting partners (HP1043_DBD/*P_hp1227_* and the HP1043N dimer) were placed at a distance of 10 Å in a guess orientation. Then the HP1043N dimer was docked on the HP1043_DBD/*P_hp1227_* complex using translations up to 3 Å and rotations up to 8° before the start of every individual simulation. A total of 500 dockings were performed, as previously described [44]. Finally, the two linkers were added to the model structure using the software MODELLER 9.18 [30]. Of the 100 models generated, the best model was selected based on the DOPE potential included in MODELLER [45]. The electrostatic potential of the HP1043 surface was calculated using DepPhi [46].

### 4.9. Electrophoretic Mobility Shift Assays

The DNA binding activity of the recombinant HP1043 in the presence of DNA groove binders or IC-substituted DNA probes was assessed by electrophoretic mobility shift assays (EMSAs). A 100-bp promoter region of *hp1227* (strain 26695 annotation; *P_hp1227_*) [47] encompassing the HP1043 binding site was amplified by PCR with oligos Php1227_EMSA_F and Php1227_EMSA_R and used as a target sequence. IC-box substituted *P_hp1227_* probe (IC-Php1227) was obtained by annealing complementary oligonucleotides (Php1227_ICbox_F and Php1227_ICbox_R) to form a double-stranded DNA. Recombinant HP1043 protein was mixed with approximately 10 ng of target promoter probe in a 20 μL reaction volume containing 10 mM Tris pH 7.5, 40 mM KCl, 100 mg/L BSA, 1 mM DTT, and 5% glycerol. For binding interference assays, DNA groove binders were added to final concentrations of 0.001, 0.01, 0.1, 1, 10, 100, 500 μM to the reaction mixtures. To ensure the specificity of EMSA assays, a 61-bp DNA fragment of the 16S rRNA gene was included as non-specific competitor DNA. Reactions were incubated at room temperature for 20 min and subsequently separated on a 6% non-denaturing polyacrylamide gel in 1× running buffer (25 mM Tris, 190 mM glycine) at 90 V, using a Mini-PROTEAN^®^ 3 Cell apparatus (BioRad, Hercules, CA, USA). DNA bands were stained with 1× ethidium bromide and visualized with a Gel Doc XR+ image analyzer (BioRad, Hercules, CA, USA).

### 4.10. Molecular Dynamics

Molecular dynamics of the HP1043 dimer bound to DNA and unbound was performed with AMBER 18 [48]. To parametrize the complex, ff14SB [49] was employed for the protein, OL15 [50] force field for DNA and water was treated as an optimal point charge. The total charge of −66 for the complex and −16 for the protein alone was balanced with Na^+^ counterions, and 89 or 56 of both Na^+^ and Cl^−^ ions were added in a water box of about 1000–650 nm^3^, respectively. Solvated complexes were minimized for 1000 steps, heated until 300 K in 100 ps followed by 50 ps of NPT equilibration. Ten simulations of 10 ns each were performed using MMPBSA, employing periodic boundary conditions. Trajectory and energetic analyses were performed using the cpptraj and MMBPSA.py tools [51]. In detail, Cα RMSD, per-residue RMSF, the distance between protein interface residues and DNA center of mass, and Cα Normal Mode Analysis were evaluated with cpptraj. MMPBSA.py was applied on 50 equally distributed frames along the joint trajectory to perform an alanine-scanning of Protein-DNA interface residues; the solvation free energy was evaluated using the modified generalized Born (GB) model [52] using 1.0, 0.8, and 4.85 values for α, β, and γ respectively; ions concentration was set at 0.150 M.

## Figures and Tables

**Figure 1 ijms-22-07848-f001:**
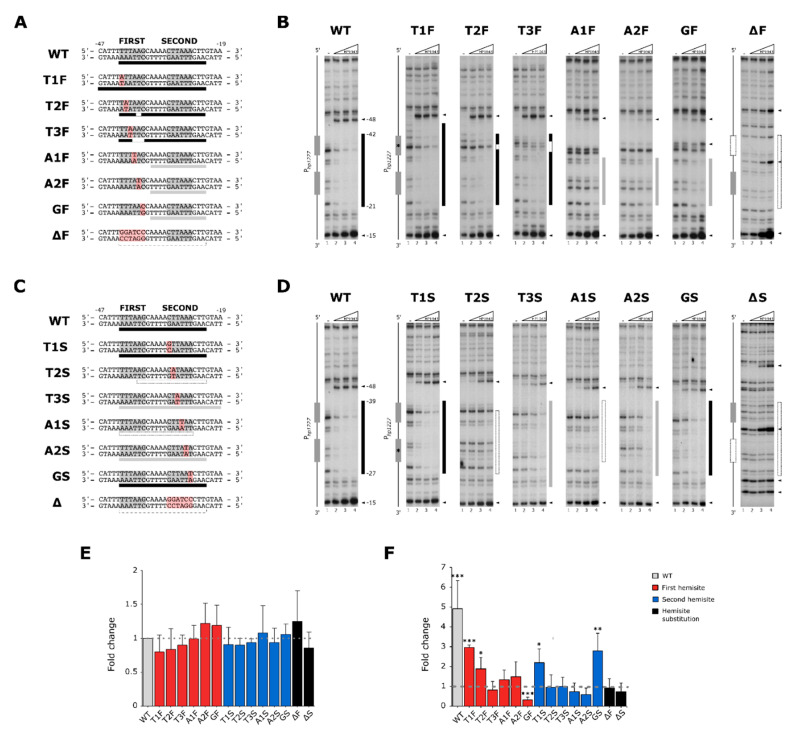
In vitro interaction of HP1043 with WT and mutant *P_hp1227_* promoter probes. Scanning mutagenesis experiments in which single bases of the consensus or one entire hemi-site were mutagenized. (**A**,**C**) Schematic representation of HP1043 binding site on the *P_hp1227_* probe used for in vitro DNA binding experiments. Hemi-sites of the HP1043 consensus are enclosed in grey boxes while mutations of the probe are highlighted in red. In (**A**–**F**), WT, wild-type probe; T1F, T2F, T3F, A1F, A2F, GF, ΔF, mutant probes as reported in panel A; T1S, T2S, T3S, A1S, A2S, GS, ΔS, mutant probes as reported in panel C. (**B**,**D**) The same DNase I footprinting experiment was used as a reference, performed in the absence (-) or with increasing amounts of dimeric HP1043 protein (0, 1.7, 6.6, and 13.3 µM, indicated as a triangle, lanes 1–4, respectively). Grey boxes mark the position of the hemi-sites of the HP1043 consensus sequence on the probe and nucleotide mutations are indicated with an asterisk. In (**A**–**D**), the length of the bars under the sequences or on the left of in vitro binding assays indicate the coverage of HP1043 protection on the DNA probe from DNase I digestion, while color indicates the strength of interaction, with black representing strong binding, grey intermediate binding, and white loss of binding. Black arrowheads indicate sites of hypersensitivity to DNase I digestion. (**E**,**F**) In vitro transcription experiments performed on *hp1227* mutant promoters controlling the *lux* operon. In (**E**), basal activity of wild-type (set to 1) and mutant promoters cloned upstream the *lux* operon in the absence of HP1043. In (**F**), transcriptional variation on the same promoters after the addition of purified HP1043 and RNAP in a 3:1 ratio. Transcriptional variation is indicated as fold-change compared to the WT in (**E**) or compared to basal levels of each condition in (**F**). In (**E**), the basal transcription level of WT is indicated as a grey dotted line. In (**F**), the grey dashed line represents the basal level of transcription of each condition. In (**F**), asterisks indicate the statistical significance of transcription levels in the presence of HP1043 compared to basal levels for each promoter, calculated by Student’s T-test, with * = *p <* 0.01, ** = *p <* 0.005, *** = *p <* 0.001. In (**E**,**F**), wild-type is represented in grey, single nucleotide mutants of the first hemi-site of the consensus in red, single nucleotide mutants of the second hemi-site in blue, while mutants in which the entire hemi-site was substituted are depicted in black.

**Figure 2 ijms-22-07848-f002:**
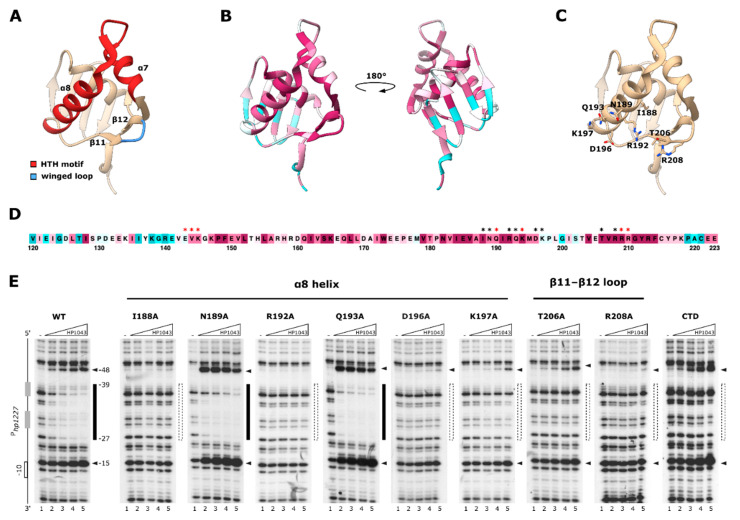
In vitro binding of wild-type and mutant HP1043 to *P_hp1227_*. (**A**–**C**) DNA binding domain (DBD) of HP1043 depicted as ribbon (PDB: 2HQR). (**A**) Functional elements of the wHTH motif. The HP1043_DBD is depicted in tan. The HTH motif is highlighted in red, while the winged loop in light blue. (**B**) Evolutionary conservation analysis was performed using default settings of the ConSurf web server on the HP1043-DBD protein sequence. Residues are colored depending on their calculated conservation scores against homologs, on a scale of nine grades ranging from the most variable positions (grade 1) colored turquoise, through intermediately conserved positions (grade 5) colored white, to the most conserved positions (grade 9) colored maroon. (**C**) residues selected for alanine scanning mutagenesis are shown as stick and colored by heteroatom. (**D**) The protein sequence of the HP1043-DBD (residues 120–223) colored as in B. Asterisks indicate residues selected for the initial mutagenesis (in black) and confirmation of the docking model (in red). The color scheme used is the same as in panel B (**E**) DNase I footprinting experiments to investigate DNA binding properties of HP1043 point mutants or in the absence of the dimerization domain (CTD mutant). Experiments were performed as in Figure 1 except for HP1043 protein concentrations tested (0, 0.7, 2, 5.9, and 17.7 μM dimeric HP1043). Increasing HP1043 concentration is indicated as a triangle, absence of HP1043 in the reaction is indicated with “-”. On the left of the autoradiographic film, a schematic representation of *P_hp1227_*, where the −10 promoter element is depicted as an open box and the hemi-sites of the HP1043 consensus sequence as grey boxes. On the right of the footprint panels, the bars indicate the region DNase I protection on the target promoter, and the color represents the strength of interaction, with black representing strong binding, grey intermediate binding, and white loss of binding. Black arrowheads indicate DNase I hypersensitive sites. WT, wild-type HP1043 protein; I188A, N189A, R192A, Q193A, D196A, K197A, T206A, R208A, amino acid substitutions in the HP1043 protein; CTD, HP1043 mutant deprived of the dimerization domain, preserving the DNA-binding domain (residues 119–223).

**Figure 3 ijms-22-07848-f003:**
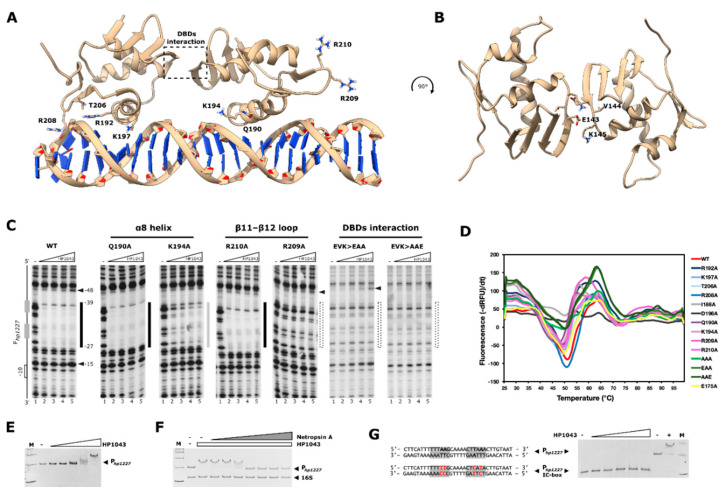
Validation of HP1043-*P_hp1227_* interaction model. (**A**) HP1043_DBD-*P_hp1227_* computational model depicted as ribbon. On the left, amino acid residues are used as restraints to guide the docking, on the right, residues are tested to validate the model. The region of putative interaction between HP1043 DBDs is highlighted. (**B**) View from above. Residues presumed to be involved in the HP1043 DBDs interaction are exposed. In (**A**,**B**), protein and DNA backbone are depicted as ribbon and colored in tan. Selected residues are depicted as stick and colored by heteroatom. (**C**) DNase I footprinting experiments were performed to validate the docking model, where residues K194 and R209, but not amino acids Q190 and R210, interact with DNA. On the right, in vitro binding experiment to investigate the putative interaction among HP1043 DBDs, by mutation of residues 143–145. Experiments were executed as described in the legend to Figure 2. WT, wild-type HP1043 protein; Q190A, K194A, R201A, R290A, EVK > EAA, EVK > AAE, amino acid substitutions in the HP1043 protein. (**D**) Representative first derivative curves of fluorescence obtained in a thermal stability shift assay performed on wild-type and mutant HP1043 proteins in the presence of SYPRO orange. The minimum of the curves corresponds to the protein melting temperature. (**E**–**G**) Electrophoretic mobility shift assays (EMSAs) to validate HP1043 interaction with minor and major grooves of the DNA. In (**E**), EMSA to estimate the concentration of HP1043 required to shift the target probe. Eighty pmol of a 100 bp probe encompassing the HP1043 binding site on the *hp1227* promoter were incubated with increasing concentrations of purified recombinant HP1043 (0, 1, 2, 4, 8, 16 μM). In (**F**), EMSA in the presence of a fixed amount of HP1043 (8 μM) and increasing concentrations of the DNA minor groove binder Netropsin (0.001, 0.01, 0.1, 1, 10, 100, 500 μM, indicated as a grey triangle). A 60 bp probe of the 16S rRNA gene was used as an internal specificity control. In (**G**), increasing amounts of HP1043 (0, 8, 12, 16, 20, 24 μM) were incubated with a *P_hp1227_* probe harboring IC-box substitutions in the sequences recognized by HP1043. To control the reaction, 8 μM HP1043 was incubated with the wild-type *P_hp1227_* DNA. IC-box substitutions are highlighted in red, hexamers of the HP1043 consensus binding sequence are enclosed in grey boxes. In (**E**–**G**), HP1043 concentration is indicated as a white triangle when used in increasing amount (in (**E**,**G**)), or as a white rectangle when used at a fixed concentration (in (**F**)). The DNA molecular weight ladder is indicated as M, the addition or lack of a component in the reaction mixture is indicated with a plus “+” or minus “-” sign, respectively.

**Figure 4 ijms-22-07848-f004:**
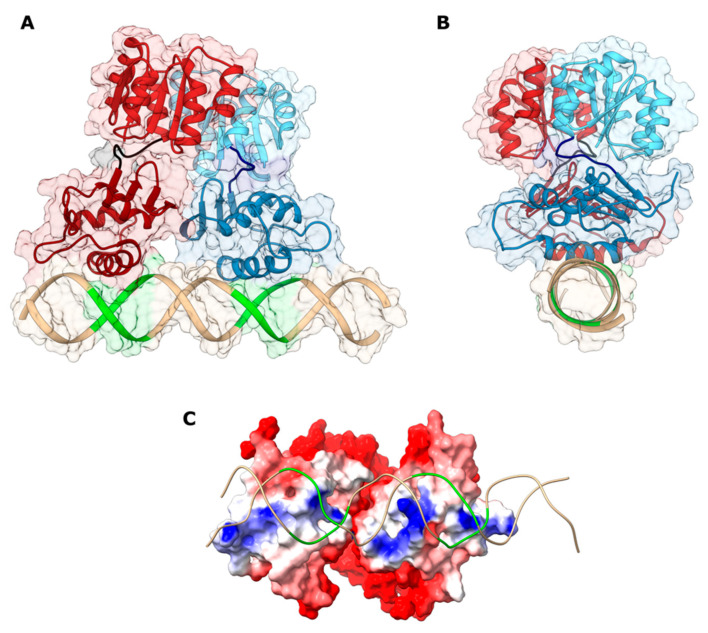
HP1043-P*_hp1227_* docking model. (**A**,**B**) Front and side view of HP1043-DNA interaction model depicted as ribbon and surface. HP1043 monomer A is colored in shades of red, monomer B in shades of light blue, and the DNA in tan, except for the active residues, reported in green. In the model, HP1043_DBDs interact with DNA in a head-to-head conformation. Helices α8 of the wHTH motif contact the DNA major grooves which in turn are broadened, while the adjacent minor grooves are narrowed and recognized by the β11–β12 wings. (**C**) Molecular surface of HP1043 oriented to expose the DNA binding region and colored according to the calculated electrostatic potential contoured from +5.0 kT/e (intense red) to −5.0 (intense blue) (where k is the Boltzmann constant, T the absolute temperature, and e the electron charge). The DNA backbone is depicted as a ribbon colored as in panel A.

**Figure 5 ijms-22-07848-f005:**
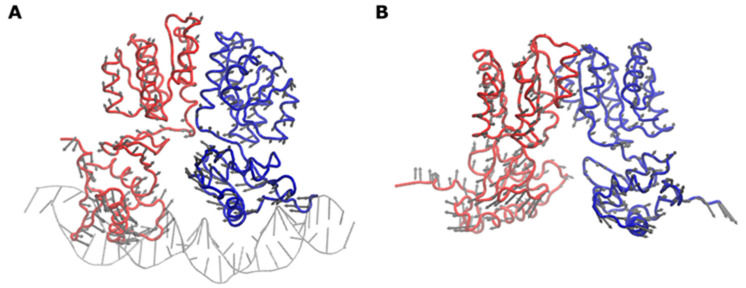
Vector field representation of the PC1 obtained from Normal model analysis of DNA-bound complex (**A**) and unbound HP1043. (**B**). Chain A is represented in red, chain B in blue and DNA is in light gray. Gray arrows show the motion quantity and direction.

**Figure 6 ijms-22-07848-f006:**
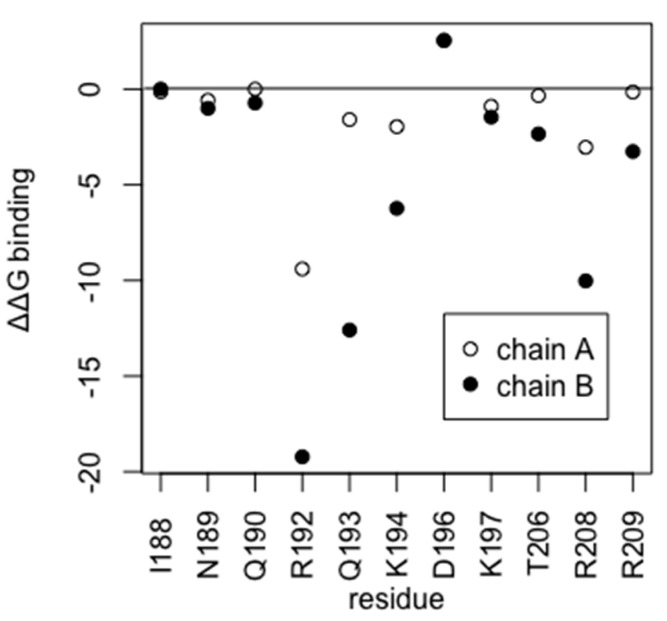
ΔΔG binding energy (kcal/mol) plot obtained from alanine scanning.

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
