# Peer review of "Definition of the Binding Architecture to a Target Promoter of HP1043, the Essential Master Regulator of Helicobacter pylori"

_ijms, 2021, doi:10.3390/ijms22157848_

Round 1

Reviewer 1 Report

The manuscript describes the binding architecture to a target promoter of the Helicobacter pylori HP1043 transcription regulator. It describes an elegantly designed study, that brings together sound experimental results with in depth in silico analysis. I believe this is a very nice paper, on a timely issue, with expected impact in its field.

Only a few minor issues could be addressed to improve, even further, the quality of the manuscript:

Line 33 – “only 17”, out of how many predicted ones? Please indicate, so that the reader may get why 17 are just a few, if that’s the case.

Line 54 – “likely not exhaustive”, because? Was ChIP-seq done in a single environmental condition, or more? Please clarify your feelings.

Line 180 – the authors acknowledge this unexpected observation: “addition of HP1043 to the GF mutant reaction caused a decrease in transcript levels”. Can you put forward an explanation for this? One thing would be to see that some promoter mutants exhibit lower basal activity altogether, but to lead to HP1043 to have a repressor behaviour is something else entirely! Would this impact the effect of this TF in natural promoters having this nucleotide sequences, turning into a repressor? Please discuss this.

Figure 6 – “chain” instead of “chian”

Author Response

Point-by-point reply (in red)

Line 33 “only 17”, out of how many predicted ones? Please indicate, so that the reader may get why 17
are just a few, if that’s the case.

Line 33 “only 17, out of approximately 1,500 predicted open reding frames (ORFs)...

Line 54 “likely not exhaustive”, because? Was ChIP-seq done in a single environmental condition, or
more? Please clarify your feelings.

Line 55 “likely not exhaustive, because of a stringent peak-calling analysis applied to identify high
confidence candidates”,

Line 180 the authors acknowledge this unexpected observation: “addition of HP1043 to the GF mutant
reaction caused a decrease in transcript levels”. Can you put forward an explanation for this? One thing
would be to see that some promoter mutants exhibit lower basal activity altogether, but to lead to HP1043
to have a repressor behaviour is something else entirely! Would this impact the effect of this TF in natural
promoters having this nucleotide sequences, turning into a repressor? Please discuss this.

Line 182- “addition of HP1043 to the GF mutant reaction caused a decrease in transcript levels”. This latter
observation, although not further investigated, could be explained as an altered HP1043 binding to the GF
mutant promoter which interferes with the activity of the RNA polymerase. In support of this hypothesis is
the appearance of a novel hypertensive site to DNase I digestion in position -40 of the GF mutant probe,
close to the -35 promoter element (Figure 1B, panel GF).

Figure 6 “chain” instead of “chian”

Figure 6 - has been corrected

Reviewer 2 Report

By means of a rigorous analysis the paper “Definition of the binding architecture……” by Zannoni et al., et al., focuses on understanding the interplay between a specific target promoter and the HP1043 regulator, an orphan response regulator which plays a relevant role in the control of  several H. pylori cellular processes. The experimental approach is very well thought out and logically alternates biochemical and computational approaches to shed light on crucial elements involved in the protein-DNA interactions. In particular, by scanning mutagenesis of the HP1043 consensus sequence and of the HP1043 DNA-binding domains the authors identify key elements required for efficient binding and use these data to generate a computational model of HP1043-promoter interaction.

This work is an elegant and effective combination of advanced methodologies and strongly contributes to decipher the molecular mechanisms adopted by HP1043 to recognize the promoter and efficiently activate its transcription.

Overall, the paper is very interesting, well presented and clearly written. The large amount of data provided fully supports the authors’ conclusions. Therefore, I strongly recommend publication in International Journal of Molecular Sciences.

Author Response

We thank the reviewer for the nice comment